# Integration of Pre-Treatment with UF/RO Membrane Process for Waste Water Recovery and Reuse in Agro-Based Pulp and Paper Industry

**DOI:** 10.3390/membranes13020199

**Published:** 2023-02-06

**Authors:** Sumit Dagar, Santosh Kumar Singh, Manoj Kumar Gupta

**Affiliations:** 1Department of Environmental Engineering, Delhi Technological University, New Delhi 110042, India; 2Environmental Management Division, Central Pulp and Paper Research Institute, Saharanpur 247001, India

**Keywords:** membrane filtration, secondary clarifier, wastewater effluent, organic pollutants, membrane fouling

## Abstract

This recent study aims to evaluate the efficacy of membrane filtration on recovery of water resource from agro-waste such as bagasse, crop-based pulp and paper mill waste. A mini pilot scale membrane system having a combination of pre-treatment filter unit (pre-filter, sediment filter and pre-carbon filter), ultra-filtration and reverse osmosis with spiral wound configuration were employed to evaluate the water reuse efficacy of effluent coming from the secondary clarifier of the conventional treatment plant of the mill. The operational conditions were optimized using Taguchi method at pH 8, temperature 32 °C, and pressure 2 bar and a flow rate of 60 l/hr. The qualities of the effluent from the secondary clarifier, and the permeate from both the combination, viz. Combination 1 (pre-treatment + ultra-filtration) and Combination 2 (pre-treatment + ultra-filtration+ reverse osmosis) were analyzed and the percentage reduction in pH, TDS, TSS, BOD, COD, Color, Lignin, Potassium and Sodium were calculated. The elimination of TDS, COD and BOD with Combination 1 was not promising (<22%). However, the installation of a RO membrane greatly reduced (>88%) the contaminants in both paper mill effluents. The obtained qualities of water from all the combinations were compared with the tolerance standard for reuse as process water. The quality of effluent from the secondary clarifier did not agree with any class of water quality. The permeate from the combination of pre-treatment and UF sufficiently reduced the TSS to reach the requirement. However, the combination of (pre-treatment + UF + RO) adequately complied with the quality standard required for reuse in the making of all grades of paper.

## 1. Introduction

Water is a pervious natural resource and the greatest gift to human civilization. India is one of the richest countries owing to possessing approximately 4% of the world’s total water resources with 1869.35 billion cubic meters (BCM) of average surface water [1], and is one of the most populous countries supporting around 17.1% of the total world population. In the last few decades, a remarkable growth in the economy has been observed in India. Overgrowing population, rapid urbanization and a booming economy along with changes in lifestyle and land use patterns have considerably impacted on India’s demand for water. Being an agrarian country, agriculture is the largest user of the total water reserve with a usage of 78% (71 BCM), followed by the domestic and industrial sectors utilizing 6% and 5% of the water, respectively [2]. Rapid demographic and economic growth has increased the water demands in all sectors. The projected industrial water demand is 161 BCM by 2050 [3]. Studies have projected the per capita water availability to be around 1191 m^3^ by 2050, making India a water-scarce country as per the International Standard of 1700 m^3^ [4].

The paper and pulp industry is one of the oldest and most economically important industries to both developing and developed nations [5]. It has eminent value in the commercial and financial expansion of the nation. Because of the crucial role of paper as an important medium of record over a long time, its production is increasing day by day. This continuous increasing demand of water among the different sectors will severely affect water intensive industries such as the paper and pulp industry. Pulp and paper manufacturing units are the third largest water-consuming industries in the country. As per the present manufacturing process available in the country, the water consumption by Indian industries is 200 m^3^–250 m^3^ per ton in agro and large pulp and paper manufacturing units which is higher than the World Bank prescribed usage of 55 m^3^ per ton of pulp and paper produced. Recycled water and effluents can significantly reduce this freshwater consumption. The waste water generated through these pulp and paper industries is comprised of suspended solids, color, inorganic compounds such as carbonates, bicarbonates, chlorides, sulphate and various toxic chemicals. The effluent has high amounts of biological oxygen demand (BOD), chemical oxygen demand (COD), lignin and its derivatives which damages the receiving water bodies. Papermaking requires proper treatment of these water fractions before reusing them. The wastewater generated must be treated using selected treatment methods such as coagulation/flocculation, flotation, biological processes, adsorption and membrane filtration before being discharged to the environment [6]. Most of the mills are still using the conventional methods that are not effective to remove the entire pollution loads. In order to sustain these industries, the use of technologies that can help to remove the maximum pollutants and make the water worth reusing can be the only solution. Recently, due to their reduced chemical utilization, short processing steps, energy saving and being able to be incorporated into the existing operating treatment plants, membrane technology has received great attention [7,8,9,10]. Various studies investigating the treatment of effluent waste water are illustrated in Table 1. A significant reduction in the pollutants such as COD, BOD, grease, fats, oils, suspended particles, etc., has been observed in these studies. However, due to the presence of a huge amount of organic pollutants and chemicals, the complexity increases for utilizing membrane filtration for paper and pulp industries. The organic materials present in the effluent deposit on the surface of the membrane and block the membrane known as fouling, which subsequently reduces the flux. The fouling of the membrane reduces its active surface area and hence decreases its efficiency; thus, it leads to a reduction in the membrane’s lifespan, higher operating costs and membrane replacement expenditure [7,11]. Significant research has been devoted to address the issue of membrane fouling focusing on the fouling characterization [11], fouling mechanisms [12], pre-treatment methods [13] and fouling prevention and cleaning regimes [14,15,16].

This current study was conducted to test the feasibility of utilizing membrane filtration viz., ultra-filtration (UF) and reverse osmosis (RO) to reduce the pollution load from the effluent of pulp and paper industry. The membranes were used in combination with a pre-treatment filter and the operating conditions of the membrane were optimized to reduce the membrane fouling and obtain the maximum possible reduction in pollution.

## 2. Materials and Methods

### 2.1. Sample Collection

In the present work, an agro-waste-based pulp and paper manufacturing mill, i.e., Bindal Paper Mills Ltd., located in Muzaffarnagar, Uttar Pradesh, India was selected for effluent collection. Grab samples of effluent were collected from the secondary outlet (final treated effluent) of the effluent treatment plant and stored by refrigerating at 4 °C and then brought to room temperature prior to experimentation.

### 2.2. Experiment Set Up

The present study was undertaken to test the efficacy of assembled effluent treatment technologies. Three membrane technologies viz., pre-filtration unit (combination of pre-filter, sediment filter and pre-carbon filter), ultra-filtration (UF) membrane and reverse osmosis (RO) membrane were set up in two different assembly combinations. The first combination was the series assembly of pre-filtration and UF membrane (Figure 1), while the second combination was the series assembly of pre-filtration, UF membrane and RO membrane (Figure 2). These combinations were employed with a diaphragm-type pump that circulated the effluent from the pre-filter set up to the UF membrane. A flow meter was connected to collect the permeate (i.e., the treated water) from the experimentation unit. The sediment filter and pre-carbon filter with a max flow rate of 100 gpm and pressure ≤8 bar and temperature of 37 °C were used. The ultra-filtration membrane made up of polyether sulphone (PES) with an area of 81.073 cm^2^, average pore size of 0.1 µ and molecular weight cut-off of 30 kDa having the capacity to withstand a pH range of 0–14, trans-membrane pressure and temperature ≤40 bars and 45 °C, respectively, was used for the experiment. A reverse osmosis membrane made of polypropylene with a capacity of 15 LPH to withstand a trans-membrane pressure of 30 Bar and temperature up to 40 °C was used.

### 2.3. Operating Conditions

The Taguchi method was used to design the experiment because of its systematic, simple and efficient approach for optimization of parameters. The Taguchi method applies fractional experimental designs called orthogonal arrays (OA) to reduce the number of experimental required to determine the optimum conditions on the results. Four different factors and three levels were selected. The selected factors and their ranges were pH: 6–8; temperature: 20–32 °C; trans-membrane pressure: 2–4 bar and flow rate 30–60 l/hr. Flux decline caused by fouling and COD rejection rate were chosen as the response parameter to evaluate the membrane fouling. For the experimental design with four factors and three levels for each factor, an L9 (3^4^) orthogonal array with eight degrees of freedom was selected. The optimal operating conditions based on Taguchi and the Grey Relation method were found to be at pH 8, temperature 32 °C, and pressure 2 bar and a flow rate of 60 l/hr.

### 2.4. Sample Analysis

The samples were analyzed for the pH, conductivity, TDS, TSS, BOD, COD, lignin, color, K^+^, Na^+^ in triplicate. All the analyses were performed as per the standard methods given provided by APHA, 2005 as provided in Table 1.

The percentage removal of various pollutants was calculated using Equation (1):(1)$$\mathrm{Rej}\;\left(\%\right)=\left(1-\frac{\mathrm{Xp}}{\mathrm{Xf}}\right)\times 100$$

Rej (%) = Percentage Rejection; X_p_ = Concentration in permeate; X_f_ = Concentration in feed.

## 3. Results and Discussion

The effluent from the secondary clarifier after biological treatment, taken from agro-based pulp and paper mills located in Saharanpur, Uttar Pradesh, producing writing and printing grades of paper, were collected for further treatment through an advanced treatment system, i.e., laboratory-scale membrane filtration system comprised of pre-filtration, ultra-filtration (UF), membrane and reverse osmosis (RO) membrane to check the water reuse potential.

### 3.1. Characterization of Effluent Secondary Clarifier

In India, most integrated paper mills use agro waste such as bagasse raw materials. A region’s climatic conditions and geography determine the physicochemical properties of wood. Untreated waste fluids might have significant biological oxygen demand (BOD), chemical oxygen demand (COD), suspended particles (primarily fibers), fatty acids, tannins, resin acids and lignin and its derivatives, depending on the raw material and the procedure involved [17]. The characteristics of the waste water effluent from the secondary clarifier are shown in Table 2. All other parameters are falling within the prescribed limits of water discharge given by CPCB.

### 3.2. Performance of Membrane Filtration Unit

Effluent samples after secondary clarifier were passed through two different combinations viz. (1). Pre-filtration and UF membrane installed in series, (2) Pre-filtration, UF Membrane and RO Membrane in series.

As shown in Figure 3, the pH of the effluent reduced to 7.19 ± 0.12 with a combination of pre-filter and ultra-filtration. On the other hand, the pH further reduced to 5.74 ± 0.51 with the addition of the RO membrane. The change in the TSS level with both combinations is shown in Figure 4, both the combinations of membrane filtration configuration were able to entirely reduce the TSS. However, the removal of TDS is not satisfactory with pre-treatment and UF, as only 7.7% (2215 ± 56.45 mg/L to 1951 ± 41.23 mg/L) of TDS could be reduced. The integration of the RO membrane with UF filtration and pre-filtration could, however, reduce 88.26% of the TDS (Figure 5). Similar results have been observed by the authors of [18,19] on wastewater from poultry slaughter houses and gray water where the application of UF reduced the TSS to approximately 98% and 100%, respectively.

In terms of COD and BOD, the reduction potential using pre-filtration and UF was 17.12% (216 ± 9.11 mg/L to 179 ± 10.76 mg/L) and 22% (27 ± 0.56 mg/L to 21 ± 0.71 mg/L), respectively. However, the addition of the RO membrane along with pre-filtration and UF appeared to reduce 100% of COD and BOD from the effluent (Figure 6 and Figure 7). These results are in agreement with the study conducted by [20] on phenolic wastewater generated from a paper mill and with the application of a combination of UF-NF/RO a reduction of 95.5% in COD has been observed. Yordanov [18] had also observed a reduction of more than 94% in BOD of the wastewater from the slaughter house with the application of ultra-filtration.

Pre-filtration and UF in series could reduce the color from 412 ± 21.10 PCU to 108 ± 10.2 PCU, i.e., by 73.78% (Figure 8). Lignin is the main component of color addition in the paper industry. In terms of lignin removal from the effluent, the pre-filtration and UF membrane fared significantly better as they were able to reduce 83% (126 ± 7.71 mg/L to 22 ± 1.01 mg/L) of the lignin although the additional RO membrane could effectively reduce the entire lignin content (100%) from the effluent (Figure 9). The application of nanofiltration by the authors of [21] on textile industry wastewater reduced the color by 100%. Similarly, the use of ultra-filtration alone rejected 98% of the synthetic dye from synthetic dye wastewater [22].

In terms of conductivity, the reduction potential using pre-filtration and UF was 31% (1365 ± 81.21 mg/L to 942.9 ± 79.37 mg/L). However, the addition of the RO membrane along with pre-filtration and UF further reduced the conductivity to 98.16% from the effluent (Figure 10).

In terms of potassium and sodium, the reduction potential using pre-filtration and UF was 14.28% (4.2 ± 0.17 mg/L to 3.6 ± 1.1 mg/L) and 19% (28 ± 2.1 mg/L to 22.70 ± 1.72 mg/L), respectively. However, the addition of the RO membrane along with pre-filtration and UF further reduced the potassium and sodium ions to 98% and 87% from the effluent (Figure 11 and Figure 12). Reverse osmosis is more effective than ultra-filtration in removal of ions such as potassium and sodium from the waste water.

The results obtained from our study are in good agreement of various previous studies to remove pollutants from waste water using both RO and UF membranes. There has been a significant decrease (*p* < 0.05) in the concentration of all pollution in both the treatment methods (Table 3). Membrane filtration at low pressure is reported to be best for bleach effluent treatment in pulp and paper mills [23]. Likewise, it has also been reported that the performance of using RO in the reduction in pulp and paper industry effluent is higher than UF and nanofiltration [24]. In addition, the authors of [25] have recommended the combined RO-UF membrane process for the treatment of wastewater.

The permeate of both the combinations was compared with the feed water characteristics required by the pulp and paper industry listed in Table 4. The effluent from the secondary clarifier cannot be reused for the production of paper since none of the parameters fall under the prescribed tolerance limits. The application of pre-filtration and ultra-filtration however reduced the pollution load but, except for TSS, none of the parameters lay in the tolerance regime. However, the permeate from the series combination of pre-filtration, ultra-filtration and the RO unit was able to achieve the tolerance limits for color, TDS and TSS to be reused in the production of all the grades of paper.

## 4. Conclusions

The purpose of the study was to evaluate the efficacy of membrane technology for the removal of pollutants from an agro residue-based paper mill secondary clarifier effluent and to check the potential for reuse of the treated wastewater for paper production. For this, a tertiary pilot system incorporating a membrane filtration unit, i.e., pre-filtration, ultra-filtration (UF), and reverse osmosis (RO) membrane were used. Two combinations of membrane filtration were used viz., Combination 1 comprised of series assembly of pre-filtration and UF whereas the RO membrane is added with pre-filtration and UF in the Combination 2. The operating conditions were optimized in order to minimize the membrane fouling and maximize the pollution rejection.

The combination of the pre-filtration filter and ultra-filtration could satisfactorily reduce the TSS, color and lignin from both the paper mill effluents. The placement of the pre-filtration filter helps to reduce the pollutant load on the membrane hence decreasing the fouling of the membrane. The pre-filtration filter also reduces the economic cost of the membrane cleaning and replacement due to fouling. However, the removal of TDS, COD and BOD using pre-treatment and ultra-filtration was not satisfactory (<30%). This may be attributed to the high concentration of low molecular weight compounds in the effluent. However, the addition of the RO membrane could significantly (>98%) reduce the pollutants from the paper mill effluent. RO membrane exhibited the highest pollutant removal ability when used in conjunction with pre-filtration and UF membrane. So, the application of pre-treatment, ultra-filtration and reverse osmosis in series could be an effective measure to help in the recovery of the vital and finite water resource and promote its reuse in the industries.

## Figures and Tables

**Figure 1 membranes-13-00199-f001:**
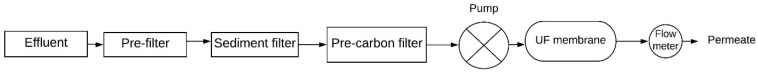
Flowchart of experiment set up (Combination 1: pre-filtration and UF membrane installed in series).

**Figure 2 membranes-13-00199-f002:**
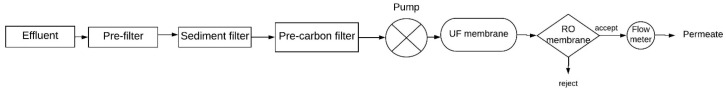
Flowchart of experiment set up (Combination 2: pre-filtration, UF membrane and RO membrane in series).

**Figure 3 membranes-13-00199-f003:**
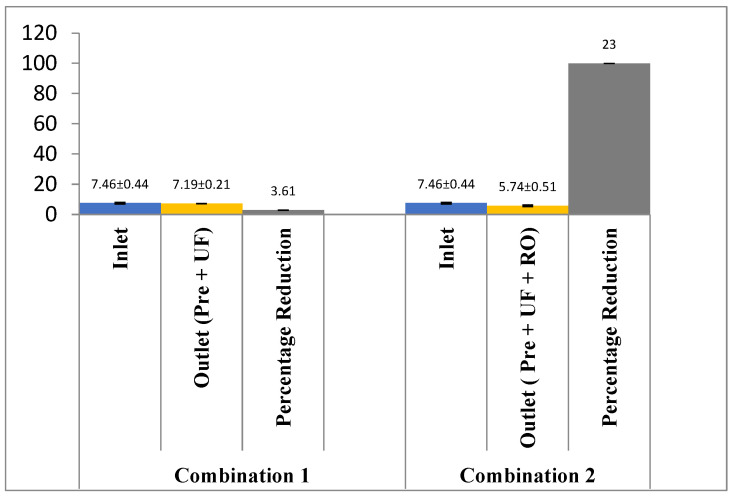
Effect of different treatment setups on the pH rejection.

**Figure 4 membranes-13-00199-f004:**
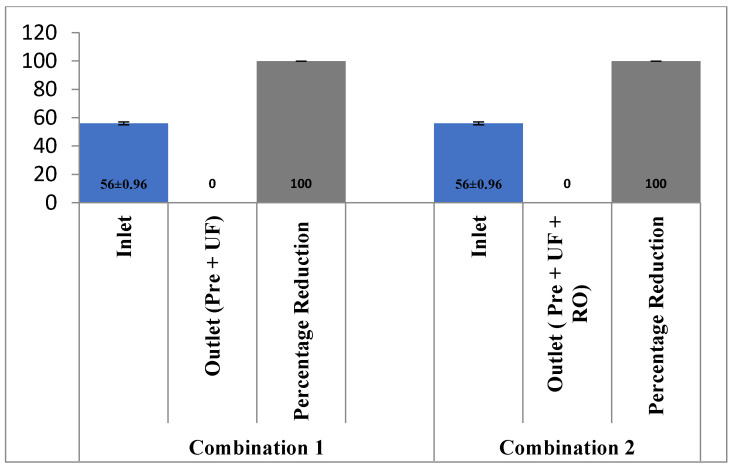
Effect of different treatment setups on the TSS rejection.

**Figure 5 membranes-13-00199-f005:**
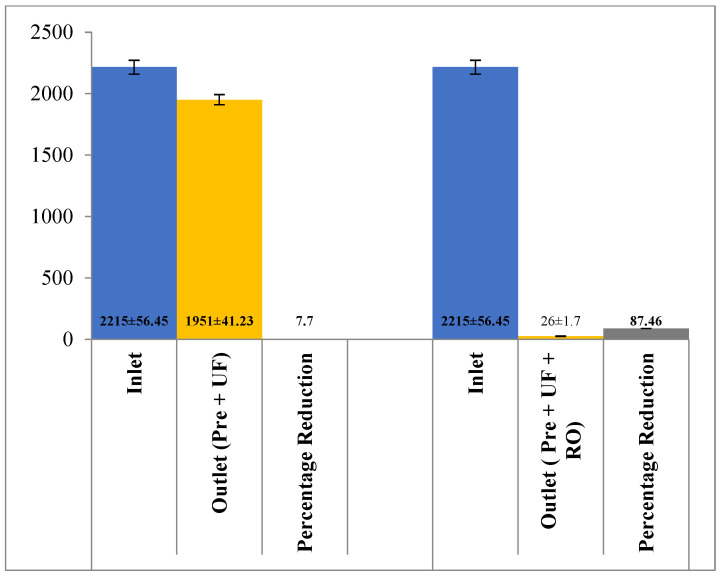
Effect of different treatment setups on the TDS rejection.

**Figure 6 membranes-13-00199-f006:**
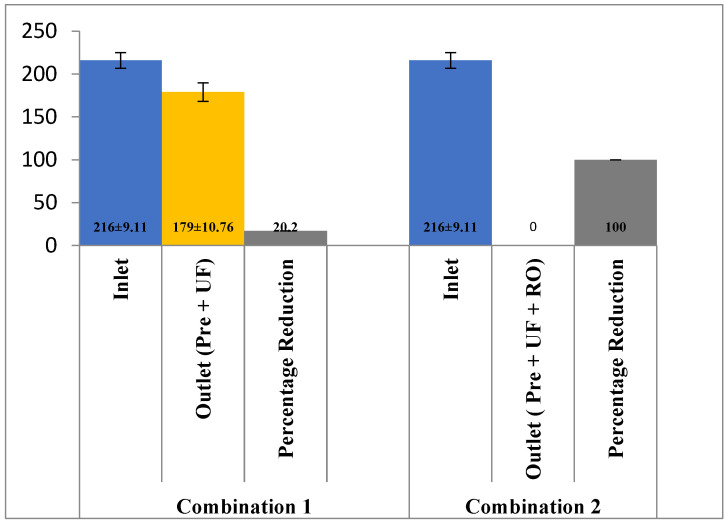
Effect of different treatment setups on the COD rejection.

**Figure 7 membranes-13-00199-f007:**
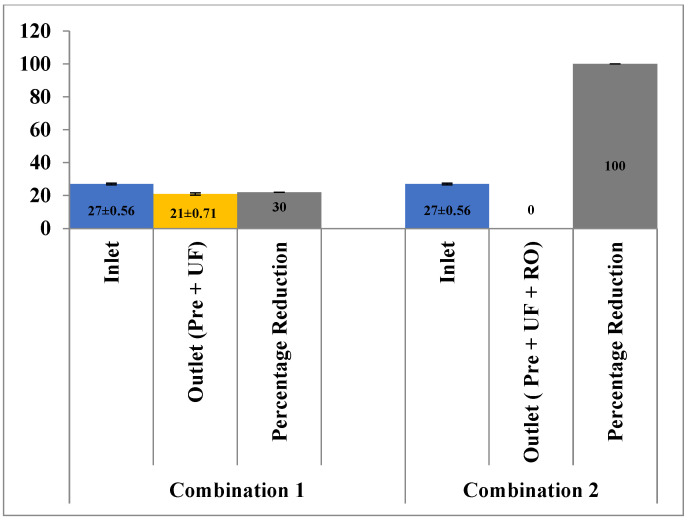
Effect of different treatment setups on the BOD rejection.

**Figure 8 membranes-13-00199-f008:**
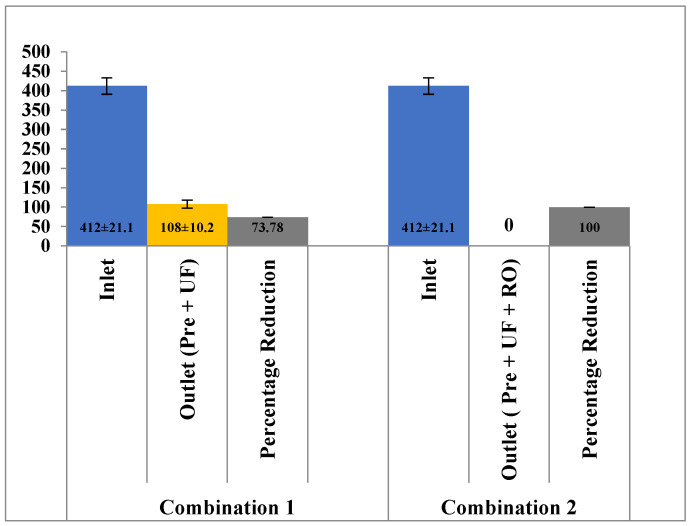
Effect of different treatment setups on the color rejection.

**Figure 9 membranes-13-00199-f009:**
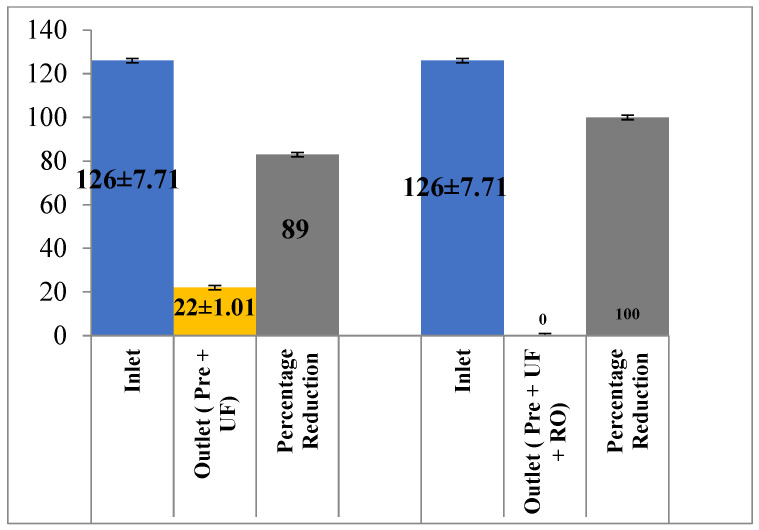
Effect of different treatment setups on the lignin removal.

**Figure 10 membranes-13-00199-f010:**
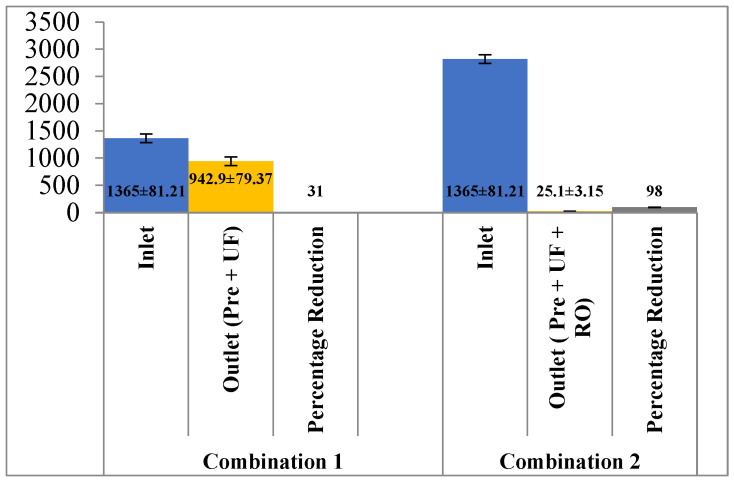
Effect of different treatment setups on the conductivity.

**Figure 11 membranes-13-00199-f011:**
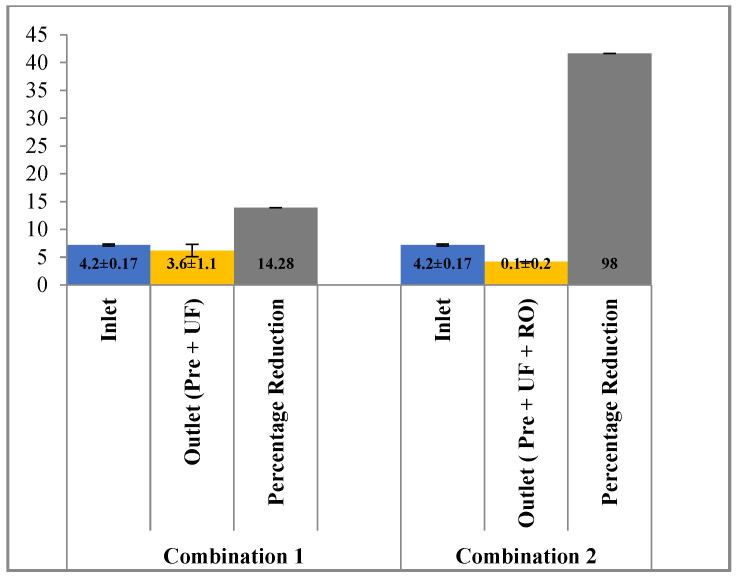
Effect of different treatment setups on removal of potassium.

**Figure 12 membranes-13-00199-f012:**
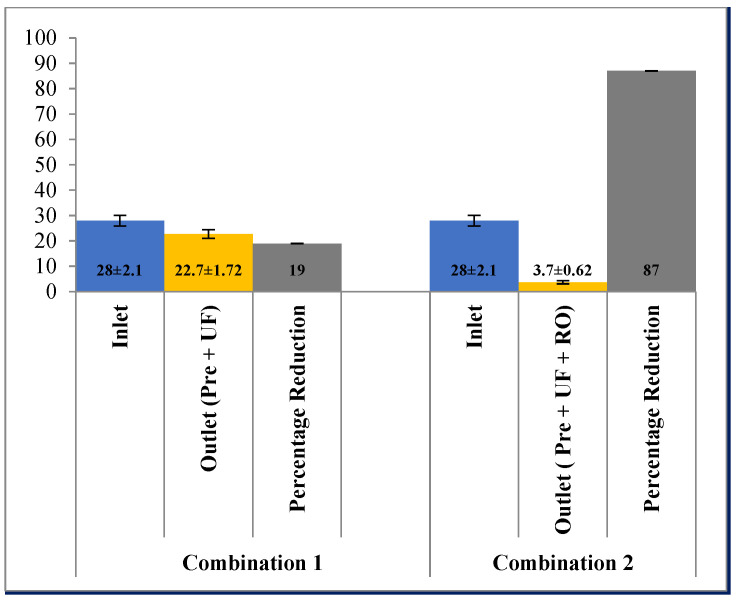
Effect of different treatment setups on removal of sodium.

**Table 1 membranes-13-00199-t001:** Physico-chemical parameters for analysis.

S.No	Physico-Chemical Parameters	Method Used
1	pH	Potentiometer
2	Chemical oxygen demand (COD)	Open efflux (Potassium Dichromate Method)
3	Biochemical Oxygen demand (BOD)	5 Days Incubation at 25 °C
4	Total dissolved solids (TDS)	IS: 3025 (Part 16)
5	Total suspended solids (TSS)	Gravimetric Method
6	Color	Spectrophotometric Method
7	Lignin	APHA method 5550 B
8	Electrical Conductivity	Conductivity Meter
9	Sodium and Potassium	Flame Photometer

**Table 2 membranes-13-00199-t002:** Physico-chemical properties of effluent from Bindal paper mill.

Parameter	Values	CPCB Limits
pH	7.46 ± 0.44	5.5–9.0
TSS mg/L	56 ± 0.96	100
TDS, mg/L	2215 ± 56.45	2100
COD, mg/L	216 ± 9.11	250
BOD, mg/L	27 ± 0.56	30
Color, PCU	412 ± 21.1	500
Lignin, mg/L	126 ± 7.71	-
Conductivity, µS/cm	2630 ± 81.21	-
Sodium (Na), mg/L	49 ± 2.1	
Potassium (K), mg/L	7.2 ± 0.17	-

**Table 3 membranes-13-00199-t003:** Change in the various water parameters with different treatment membrane setups.

	**TSS (mg/L)**	**TDS** **(mg/L)**	**COD** **(mg/L)**	**BOD (mg/L)**	**Color** **(PCU)**	**Lignin (mg/L)**	**pH**	**Conductivity (µs)**	**Potassium** **(mg/L)**	**Na** **(mg/L)**
**Inlet (Secondary Clarifier)**	56 ± 0.96	2215 ± 88.26	216 ± 9.11	27 ± 0.56	412 ± 21.1	126 ± 7.71	7.46 ± 0.44	1365 ± 81.21	4.2 ± 0.17	28 ± 2.1
**Outlet (** **Pre-treatment** **+ UF)**	0 *	1951 ± 56.45 *	179 ± 10.76 *	21 ± 0.71 *	108 ± 10.2 *	22 ± 1.01 *	7.19 ± 0.12 *	942.9 ± 79.37 *	3.6 ± 1.1 *	22.7 ± 1.72 *
**Outlet (** **Pre-treatment** **+ UF + RO)**	0 *	26 ± 1.70 *	0 *	0 *	0 *	0 *	5.74 ± 0 *	25.1 ± 3.15 *	0.1 ± 0 *	3.7 ± 0.62 *

* Significant at *p* < 0.05, Two-tailed independent *t*-test.

**Table 4 membranes-13-00199-t004:** Water Quality requirements for paper and pulp industries in India.

	Effluent from Different Processes			Tolerance for Water for Pulp and Paper Industry (BIS)			
Water Quality Parameter	Effluent from Secondary Clarifier	Permeate from Pre-Filtration and Ultra- Filtration Unit	Permeate from Pre-Filtration, Ultra-Filtration and RO Unit	Ground Wood Paper	Kraft Paper Bleached	Soda and Sulfite Paper	High Grade Paper
**Color**	615	176	0	20	15	10	5
**TDS**	1933	1784	0	500	300	300	300
**TSS**	36	0	0	25	25	25	10
**COD**	183	146	0	NS	NS	NS	NS

NS—Not specified.

## Data Availability

Not applicable.

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
