# Peer review of "Integration of Pre-Treatment with UF/RO Membrane Process for Waste Water Recovery and Reuse in Agro-Based Pulp and Paper Industry"

_membranes, 2023, doi:10.3390/membranes13020199_

Round 1

Reviewer 1 Report

Dear authors,

your research topic is very interesting and deals with one of the today's major issues concerning water pollution and reducing industry waste. You were presenting results of different methods for waste water recovery and it is my opinion that it should be presented more clearly and scientifically. First, text should be corrected grammatically, some capital letters are unnecessary, unique font etc. Please ensure that each abbreviation in text has it's full name when first mentioned.

Further, methods used for sample analysis should be described individually with according chemicals that were used, or al least provide references from where methods were used.

Regarding figures and tables, results in figures are the same as in Table 2, which is unnecessary and results should not be repeated. It is better to leave the table, and provide some statistical analysis for results.

However, this manuscript requires some major corrections, even some additional research to complete and improve the current results. After that, it could be reconsidered for publishing in this Special issue.

Author Response

  1. The manuscript is corrected for grammatical errors. Each abbreviation in text has expanded when mentioned first time.
  2. Since it is very difficult to describe the complete method used for analysis, so the name of the method has been provided and reference has been added.
  3. T-test has been applied to check the level of significance and the same has been represented in table 2.
  4. Kindly suggest what additional research that can be done to improve the current results.

Reviewer 2 Report

The work entitled “Integration of Pre-treatment with UF/RO membrane process for waste water recovery and reuse in agro based pulp and paper industry” is an interesting study on the use of membrane technologies for the production of reclaimed water for industrial use. The work is interesting and well-written. Some specific comments are below:

1) Keywords should be improved as some of the words used are already on the title.

2) On topic 2.2 authors should add a schematic image of the pilot plant.

3) Topic 2.3 is really short. Or authors give more information about operational conditions, or they should add this information on topic 2.2.

4) On topic 2.4 authors must include more information about the equipment used.

Author Response

  1. The keywords have been improved as per your suggestion.
  2. The schematic image of the pilot plant has been inserted in the manuscript as figure 1 and 2.
  3. The Optimization of operational conditions mentioned in 2.3 has been improvised.
  4. The methods used for analysis has been provided as table 1 in the revised manuscript.

Round 2

Reviewer 1 Report

Authors have kindly corrected the manuscript according to my comments and therefore it is my opinion that it could be published in this special issue.